# Bacteriophages as Potential Clinical Immune Modulators

**DOI:** 10.3390/microorganisms11092222

**Published:** 2023-09-01

**Authors:** Estêvão Brasiliense de Souza, Aguinaldo Roberto Pinto, Gislaine Fongaro

**Affiliations:** 1Laboratory of Applied Immunology, Department of Microbiology, Immunology and Parasitology, Federal University of Santa Catarina, Florianópolis 88040-900, SC, Brazil; aguinaldo.pinto@ufsc.br; 2Laboratory of Applied Virology, Department of Microbiology, Immunology and Parasitology, Federal University of Santa Catarina, Florianópolis 88040-900, SC, Brazil

**Keywords:** bacteriophages, inflammation, immunology, immune modulation, anti-inflammatory

## Abstract

Bacteriophages (phages for short) are bacteria-specific viruses that have been drawing attention when it comes to countering the ever-growing antibiotic bacterial resistance, and are being seen as one of the most promising technologies against multi-antibiotic-resistant bacteria. Although bacteriophages are commonly regarded only as anti-bacterial objects unable to directly interact with eukaryotic cell metabolism, an increasing quantity of evidence has indicated that bacteriophages can directly affect cells bacteria in both in vitro and in vivo applications, influencing the behavior of tissues and immune systems. In sight of this new range of applications, several authors have expressed enthusiasm in phage therapy as direct modulators of eukaryotic cells for clinical usage, highlighting the need for further investigations covering the pharmacology of these new “eukaryotic-viruses”, as even harmful interactions with eukaryotic cells were detected after phage therapy. The present review aims to cover and highlight mechanisms through which bacteriophages may interact with immune cells, analyzing potential clinical applications and obstacles presented in the use of bacteriophages as anti-inflammatory tools.

## 1. Introduction

Bacteriophages are bacterial viruses that have been extensively explored for their antibacterial properties due to their capacity to infect and destroy virtually all pathogenic bacteria by co-evolving with their host, disrupting even multi-antibiotic-resistant bacteria found in persistent infections [1,2]. Phage therapy has demonstrated promising clinical viability and safety of use against several illnesses and conditions caused by pathogenic bacteria, ranging from the treatment of burn wounds, urinary tract infections, diabetic foot ulcers and pulmonary infections [3,4]. It has also been noticed that, besides providing bacterial infection clearance, the administration of bacteriophages directly interferes with eukaryotic cells, affecting the mechanisms of the inflammatory response, innate immunity and even reducing the rejection of transplanted tissue [5,6,7,8,9,10]. An investigation of bacteriophage structures in the eukaryotic tissues revealed that phage therapy may directly and indirectly affect cell homeostasis, affecting their active binding sites, modulating their transcription factors and overall interfering with both intracellular and extracellular environments, extending its therapeutic value beyond being only for “anti-bacterial viruses” [11,12,13,14,15]. Bacteriophage immune modulation reveals promising prospects for the employment of phages as biological tools to treat immunological diseases. This is especially made possible due to the extensive history of bacteriophage manipulation, its high target specificity, inherent antibacterial function and cheap large-scale production using well-defined bacterial propagation methods, thus proving to be a cheap and highly flexible drug [16,17,18]. This newfound realm in bacteriophage studies has given rise to both enthusiasm about eukaryotic-focused phage therapy lines and the need to investigate potential undesirable interferences of phage therapy in eukaryotic systems [11,12,13,14,15,19,20,21].

When it comes to alterations in cell homeostasis, anti-inflammatory effects are reported to be tied to a reduction in pro-inflammatory cytokines and chemokines and inflammation-associated genes followed by phage exposure in both in vivo and in vitro assays, as well as a reduction in reactive oxygen species (ROS) production in phagocytes [5,7,22,23,24,25,26,27,28]. Besides decreasing inflammatory modulators, phage exposure also increased anti-inflammatory factors, further directing the cells to an anti-inflammatory phenotype [26,29]. This new anti-inflammatory potential of phage therapy has drawn special interest to its use against several inflammatory-associated diseases and conditions like arthritis, transplant rejection and overall bacteria-associated inflammation [5,6,25,30,31].

## 2. Decrease in Pro-Inflammatory Modulators in In Vivo and In Vitro Models

The use of phages was tied to the downregulation of several pro-inflammatory cytokines, chemokines and inflammation-associated genes in bacterial infection models. The modulation of immune factors includes the downregulation of interleukin 6 (IL-6), IL-8 and tumor necrosis factor alpha (TNF-α) (major cytokines in inflammation), IL-1β, nuclear factor-κB (NF-κB), C-X-C motif chemokine ligand 12a (CXCL12a), Toll-like receptor 4 (TLR4) activity (in which its activation stimulates pro inflammatory compounds) and the overall inhibition of T cell activation in both in vitro and in vivo models [5,22,23,24,25,26,28]. Phage application also presented upregulation of anti-inflammatory factors such as suppressor of cytokine signaling 3 (SOSC3), IL-1 receptor antagonist (IL1RN) and IL-10 [26,29]. These anti-inflammatory properties of phages were significant enough to reduce inflammation in lung and urinary infections in mice and even to reduce rejection in murine skin graft transplants [5,25,30,31].

Van Belleghem et al. conducted an extensive comparative study on the different immune influences between four *Pseudomonas aeruginosa* phages and one *Staphylococcus aureus* phage and their respective bacterial hosts to map their immunomodulatory mechanisms and to analyze how inflammatory gene modulation differs when cells are submitted to purified phage particles from different strains [26]. In the study, the phage solution was able to modulate 359 genes in human mononuclear cells while remaining unaltered by the respective bacterial hosts, resulting in a predominant anti-inflammatory activity with an upregulation of IL1RN and strong reduction in CXCL1 and CXCL5, presenting little difference among the analyzed bacteriophages [26]. In fact, the anti-inflammatory effects caused by the presence of bacteriophages are well known to be present in persistent infections, in which bacteriophages provide immune evasive properties to their bacterial host by reducing immune cell recruitment molecules, thus increasing the survivability of the bacteria by reducing inflammation through a commensal interaction [32]. This behavior was observed in the *P. aeruginosa*-associated phage, Pf4, where its presence in human macrophage cells was found to reduce lipopolysaccharide (LPS)-induced inflammation, decreasing the production of TNF-α, GROα/CXCL1, CXCL5, IL-6, IL-1α, IL-1β and granulocyte–macrophage colony-stimulating factor (GM-CSF), as well as reducing neutrophil migration in mice via a TLR3 and interferon-alfa/beta receptor (IFNAR)-dependent interference (Figure 1) [33].

## 3. Phage Anti-Inflammatory Activity Is Not Restricted to Bacterial Elimination

Since bacteriophage application is known to reduce inflammation in bacterial infections, phage-mediated anti-inflammatory findings were speculated to occur due to a decrease in bacterial numbers, lowering the inflammatory modulators by decreasing the infection that activated these factors in the first place [22,25,31,34]. Among other studies, such an assumption was made by Wang et al., who noticed a reduction in inflammatory cytokines after phage application in *S. aureus* infections (both in vivo and in vitro), while the same could not be observed when phages were introduced in bacteria-absent mice [35]. However, studies revealed that the anti-inflammatory response can be caused by the direct interference of the bacteriophage structure in immune cells, as a reduction in inflammatory modulators was observed in mononuclear cells treated with highly purified phage inocula (that is, a phage solution without the presence of live bacteria or bacterial fragments) [26]. Similar studies observed anti-inflammatory effects when no bacterial model was utilized. Cafora et al. revealed a decrease in the expression of IL-1β, IL-6, IL-8 and CXCL12a in zebrafish injected with a phage cocktail [28]. *P. aeruginosa* phage, Pf4, also managed to decrease TNF-α production when only LPS was applied in murine phagocytes and human monocytes (Figure 1) [32]. Likewise, a study by Górski et al. exhibited the reduction in NF-κB activity and inhibition of T-cell activation and proliferation after phage treatment in human cells infected by herpes simplex virus type 1 (HSV-1). The same study also managed to reduce the inflammation and rejection of skin transplants in mice after phage application, reducing the infiltration of mononuclear cells in the transplant [5]. This direct interaction with immune cells was also seen in murine models of autoimmune diseases, in which endotoxin-purified T4 phages were able to decrease the severity of murine collagen-induced arthritis after T4 intraperitoneal injection [6].

## 4. Reduction in Inflammation beyond LPS Entrapment

While pro-inflammatory factors were related to bacteriophage interference, this raises the question of whether the apparent direct anti-inflammatory activity of phages in immune cells might be due to phage binding to LPS molecules and not to a direct interaction with the cell mechanisms. There might be relevance to this inquiry as Miernikiewicz et al. revealed that phage particles are able to bind and thus decrease the inflammation caused by free LPS [36]. Such interaction is relevant in the discovery of the original source of the anti-inflammatory effect, since LPS is a common component in phage inocula, as it may arise from improper phage purification and cause inflammatory processes [37]. Treatment with the purified T4 gp12 tail-protein in mice was able to bind to LPS molecules and it significantly decreased inflammation, reducing inflammatory cytokines IL-1α and IL-6 while decreasing leukocyte infiltration in inflamed organs [36]. The direct binding of gp12 to LPS was confirmed in vitro, which demonstrated that gp12 was able to cover isolated LPS molecules [36]. Another study found that phage stimulation reduced LPS-induced IL-1β production in murine peritoneal macrophages, although no conclusive remarks on the phage-cell mechanisms could be made [38]. This interaction with LPS was suggested as a possible explanation in studies where bacterial lysate produced a high inflammation level whereas the addition of phage particles caused no visible inflammation, suggesting the reduction in bacterial-provoked inflammation was caused by bacteriophages in a similar interaction exerted by the T4 gp12 protein in LPS particles [6,36]. Some authors, however, have proposed that LPS–phage interaction may only be part of the anti-inflammatory effect, as phages were able to retain anti-inflammatory effects even when the LPS concentration was too high to be “captured” by the phage inoculum [39]. Furthermore, despite the presence of LPS, phage particles were able to directly bind and possibly interact with leukocytes, thus being implied as the true reason for immunological modulation [39]. Alternatively, the inhibition of LPS provoking inflammation may originate as a consequence of an antiviral response provoked by the phage genome. Sweere et al. reported that the recognition of bacteriophage RNA through TLR3 triggered type 1 interferon production and a consequential TNF decrement, exemplifying another anti-inflammatory effect unrelated to LPS binding [32]. 

## 5. Decrease in ROS Production in Phagocytic Cells

Bacteriophage presence was found to help the phagocytosis rate of invading bacteria while also reducing the ROS production by-product, stemming from phagocytosis activity, being able to infect the bacteria while still inside the cell [7,8]. While the studies imply that bacteriophages were not able to directly interfere with intracellular killing after phagocytosis, such indirect modulation was able to protect phagocytic cells from cytotoxic damage elicited by ROS production, as ROS and cytotoxic damage in phagocytes were significantly reduced when bacteria were pre-absorbed by phage particles [7,27]. Such interplay may be beneficial for phagocyte integrity, as although ROS within phagocytic cells play a role in pathogen control, excessive ROS accumulation may lead to cellular damage [27,40,41]. In this context, phages have exhibited a crucial role in mitigating ROS-induced damage by not only reducing the harmful effects of ROS, but also effectively clearing bacterial infections [27,39]. However, not all mechanisms related to internalized bacteria killing and ROS attenuation by bacteriophage interactions are fully defined, with authors implying that ROS reduction might be due to either the reduction in pro-inflammatory particles through phage binding to LPS molecules or some other unknown direct effect in phagocytes [39]. Such an assumption was made in a study by Miedzybrodzki et al., in which ROS reduction was higher than phage particles could bind, and therefore LPS molecules were inhibited, inferring that some other factor played a role in ROS reduction [39].

## 6. Known and Potential Mechanisms behind Bacteriophage Anti-Inflammatory Effects

Since bacteriophages are exclusive bacterial viruses which are not able to replicate inside eukaryotic cells, the seemingly direct interaction with animal cell metabolism and interference with immune signals appears as an unexpected behavior for a bacterial-targeted virus, thus drawing interest among researchers regarding the specific mechanisms behind bacteriophage–eukaryote interactions. A possible explanation as to why a virus seemingly exclusive to bacteria may exert anti-inflammatory effects might be the direct interference by phage proteins in cell receptors, such as the binding to the cell integrins and toll-like receptors [10,28]. T4 phages appear to interfere with cell metabolism by binding to β3-integrins through the KGD (Lys-Gly-Asp) motif present in the gp24 proteins of the capsid, which in turn could compete and potentially interfere with the β3-integrin regulatory activity of platelets, monocytes, lymphocytes, carcinoma cells and even in cell senescence, interfering with their natural inflammatory profile (Figure 1) [10,42,43,44,45,46,47]. The binding of phages to platelets is hypothesized to be able to reduce platelet-mediated inflammation, as platelets can support inflammation by increasing endothelial permeability, recruiting surrounding leukocytes, and by releasing inflammatory cytokines [48,49,50,51]. Also, the cluster of differentiation 40 ligands (CD40L, CD154), a crucial protein in lymphocyte activation, possesses a KGD motif which allows the protein to activate platelets via β3-integrins [52]. The inhibition of the CD40L is associated with reduced inflammation and development of autoimmune diseases [53]. Therefore, it has been hypothesized that phages that possess the motif KGD could compete with CD40L and exert an anti-inflammatory effect (Figure 1) [10]. Similar to T4 protein interference in cell inflammatory mechanisms, a report from Cafora et al. demonstrated that a *P. aeruginosa* phage cocktail decreased pro-inflammatory signals in zebrafish through the interference of the virus proteins through a myeloid differentiation factor (MyD88)-dependent TLR pathway (Figure 1) [28].

Alternatively, bacteriophages may also exert an anti-inflammatory effect through the interference of their genetic material. The recognition of the phage RNA by TLR3 was found to reduce TNF-α expression in murine bone-marrow-derived dendritic cells through the activation of type 1 interferon production, directing the cells to an antiviral phenotype instead of an antibacterial-directed inflammation (Figure 1) [32]. In fact, the induction of antiviral cytokines, such as interferons alpha (IFN-ɑ) and gamma (IFN-γ), and interleukin-12 (IL-12) is widely found after bacteriophage application in both in vivo and in vitro reports [32,54]. Unsurprisingly, the bacteriophage antiviral response (and thus the consequential reduction in TNF-ɑ and other bacterial-focused immune responses through a similar mechanism) is to be expected, since bacteriophages are recognized as invading viruses by the body along with mammalian viruses.

However, although immunomodulatory effects were present both due to the interactions with phage proteins and genetic material, it is important to highlight that both effects may not be present in all phage strains, whether the effect is due to surface protein epitopes or nucleic acids, being directly dependent on the specificities of each virus. Such divergence was seen by Sweere et al. and Cafora et al. [28,32]. The former reported that the anti-inflammatory effect originated directly from the recognition of the phage genome by TLR3, while the latter found that a *P. aeruginosa* phage cocktail could only decrease pro-inflammatory signals in zebrafish when the virus protein coat remained intact, causing no immune modulation when only the phage genome was applied, thus implying that the anti-inflammatory effect was due to the interference of the virus proteins. Similar effects in other toll-like mechanisms were observed post phage application, with the downregulation of TLR4 and upregulation of Toll-like receptor 10, the latter known to induce anti-inflammatory effects upon activation, although the mechanisms of phage structure behind the interference of these receptors were not defined (Figure 1) [26,55].

## 7. Increase in Pro-Inflammatory Signals

While the presence of anti-inflammatory effects after bacteriophage application was observed, the opposite result was also detected, with studies reporting a reduction in anti-inflammatory modulators and an increase in inflammatory signals. Phage therapy was tied to an increase in pro-inflammatory cytokines TNF-α and IL-6 and the C-C motif chemokine ligand 2 (CCL2/MCP-1) chemokine [56]. Similarly, Van Belleghem et al. found upregulated inflammatory cytokines and chemokines after phage application, namely CXCL1, CXCL5, IL1-α, IL1-β, IL-6 and TNF-α; however, the effect was reported as predominantly anti-inflammatory due to a reduction in IL-1RN production [26]. Park et al. compared the inflammatory cytokine profile between phage and murine norovirus (MNV) in mice treated through oral administration, finding an increase in IL-17α in phage-fed animals [57]. Although MNV inoculation significantly increased IL-1α, IL-1β, IL-2, IL-10, IL-12, IL-17α, IFN-γ, TNF-α, granulocyte colony-stimulating factor (G-CSF) and granulocyte-macrophage colony-stimulating factor (GM-CSF), phage administration was still only related to a minimal inflammatory response [57]. Phage usage was also observed to reduce the anti-inflammatory cytokine IL-10, although the same virus also reduced IL-1β and TNF-α [23,26]. Gogokhia et al. found that different phage strains could stimulate inflammatory cytokines in bacteria-depleted mice, revealing a significant induction of IFN-γ after purified phage DNA was introduced to dendritic cells [9]. Phage treatment also increased the local concentration of CD4+ T cells in germ-free mice, achieving comparable levels in mice infected only with bacteria [9]. Such an inflammatory profile remained even when the recognition of LPS was inhibited, implying that the effect was, once again, not due to LPS contamination but hence due to the immunoreactivity of the phage genetic material, as DNA-free phages were not able to induce an immune response [9]. 

## 8. Bacteriophage Immune Modulation Variables

The presence of both inflammatory and anti-inflammatory modulators post bacteriophage interaction gives rise to the hypothesis that bacteriophages may induce a wide range of inflammatory responses which seem to be predominantly pro-inflammatory or anti-inflammatory, depending on the final immunological balance. This divergence was present in a report by Van Belleghem et al., where, although phage-stimulated pro-inflammatory factors such as IL-6 and TNF-α were present, the final balance was defined as anti-inflammatory, due to a drastic reduction in IL1-RN [26]. Another factor that needs to be taken into consideration is that pro-inflammatory and anti-inflammatory responses vary between phage strains and viral concentration, with bacteriophages derived from the same host presenting distinct cytokine stimulation based on the specific strain and viral titer. Different *Escherichia coli* phage strains induced divergent IL-6, IL-10 and TNF-α responses in human colorectal adenocarcinoma cells (HT-29), demonstrating that viruses of the same family and host strain may possess different immunogenicity degrees [58]. Similar results were observed in *P. aeruginosa* phages, where the stimulation with certain phage strains induced IL-6 and TNF-α production in cell lines, while in other strains no inflammatory effect was observed [59]. A hypothesis formulated by similar results implied that the increase in inflammatory modulators in in vivo phage therapy may be caused by phage-specific antibodies, in which bacteriophages may generate an inflammatory response to themselves through adaptive immunity, causing phage application to promote an inflammatory response when recognized by the acquired immunity of the organism [9,60]. Although phage particles are often described as non-immunogenic, several studies in both human and animal models have reported the presence of anti-phage antibodies during or after phage treatment, with some even impairing the therapy’s success [60,61,62,63,64]. The development of anti-phage adaptive immunity is largely influenced by the phage strain immunogenicity, application route, phage dosage and exposure time, determining both whether anti-phage responses may occur or not and the resultant immune modulation of the phage therapy [64,65,66]. The influence of the application route in phage therapy is determined by the immune reactivity of the administered route and by how far the local area of application is from the targeted tissue, with therapy effectiveness being inversely proportional to the number of obstacles between the target and the phage inoculum [19,64]. Higher phage dosages in phage therapy were found to proportionate a pro-inflammatory effect, since certain inflammatory cytokines were only detected in high-phage titers (10^9^ PFU/well), which is a consonant result with other studies relating higher phage titers with an increased immunogenic response [66,67]. Although factors related to phage exposure were found to dictate phage adaptive immunity and partially explain divergences in the immunoreactivity of phage therapy, the same factors were also observed as influencing in vitro studies, with long application times influencing the final inflammatory profile. Extended exposure time was found to change the pro-inflammatory activity of the M13 bacteriophage in murine macrophage cell lines (RAW) to a predominant anti-inflammatory effect in the report by Sun et al [68]. In this study, Sun et al. described that M13 bacteriophages provided an initial pro-inflammatory signal in the initial 24 h of exposure via an increase in TNF-α, whereas after extended exposure (7–10 days), the virus reduced TNF-α and IL-6 production while increasing IL-10 [68]. 

The presence of both anti-inflammatory and pro-inflammatory activity within a phage therapy may also be a consequence of the range of stimulations present on the phage itself. Bacteriophages that were able to reduce the inflammatory cascade caused by pro-inflammatory molecules, such as LPS, may also induce a low level of pro-inflammatory cytokines and chemokines due to the immune recognition of the “invasive” viral structure, as observed in the report by Zhang et al. [69]. In this study, Zhang et al. reported that *S. aureus* bacteriophages were able to reduce TNF-α, IL-1β, IL-6 and IL-8 in bovine mammary epithelial cells (MAC-T) stimulated with LPS while inducing a low level of the inflammatory mediators when the viruses were added without LPS presence; this implies that although the presence of the bacteriophage, by itself, was pro-inflammatory, the resulting effect against a highly inflammatory body like LPS resulted in a mainly anti-inflammatory effect [69]. This may indicate that the denomination of bacteriophages as pro- or anti-inflammatory objects is dependent on its surrounding factors, and that it is possible to presume that bacteriophages are not anti-inflammatory objects by themselves, but instead possess anti-inflammatory activity. Such factors may explain the variability of the immune modulation after phage therapy, and highlight the need to account for phage diversity and the application method for the use of bacteriophages as anti-inflammatory objects (Figure 2). Considering all divergences previously stated, it is also possible that phages that possess anti-inflammatory properties may express an opposite effect when influenced by the application route, dosage and exposure time, with the same also applying to phages that were pro-inflammatory, although more data are still required.

## 9. Conclusions

Phages can directly act on eukaryotic tissues and influence the activity of immune cells; however, not all mechanisms of these interactions have been fully mapped yet. Although positive results were obtained, due to the presence of both adverse and non-responding effects in phage application, i.e., whether the therapy was hindered by anti-phage antibodies or overall problems with the product’s pharmacology, one can understand why antibiotics were widely adopted regarding antibacterial action, while phage therapy became mostly restricted to research usage. The undefined pharmacology in phage products together with the lack of consensus on administration methods and the unknown patient immunoreactivity to the viral product may lead to serious divergences in therapy effectiveness. 

Moreover, although phage pharmacology is still a largely unexplored field, bacteriophages’ newfound potentials of immunomodulation show great promise as potential anti-inflammatory therapeutics, with distinct viral strains demonstrating potential against inflammation and autoimmune diseases. Bacteriophages are easily propagated and possess a highly programmable structure, providing a low-cost biological tool that can be modified to meet the necessary pharmacology for clinical usage. Additionally, the high diversity of bacteriophages allows for the bioprospecting of wild viral strains that may already present the targeted immunomodulatory properties together with the required pharmacology. Although the immunomodulation of bacteriophages has been explored in in vivo and in vitro studies, there are still no clinical trials on the use of bacteriophages as potential anti-inflammatories. A new field of study is yet to be explored to investigate and evaluate the use of bacteriophages as immunomodulatory objects, aiming to analyze the potential employment of the immunomodulatory properties of phages as a new therapeutic avenue against multiple animal and human diseases.

## Figures and Tables

**Figure 1 microorganisms-11-02222-f001:**
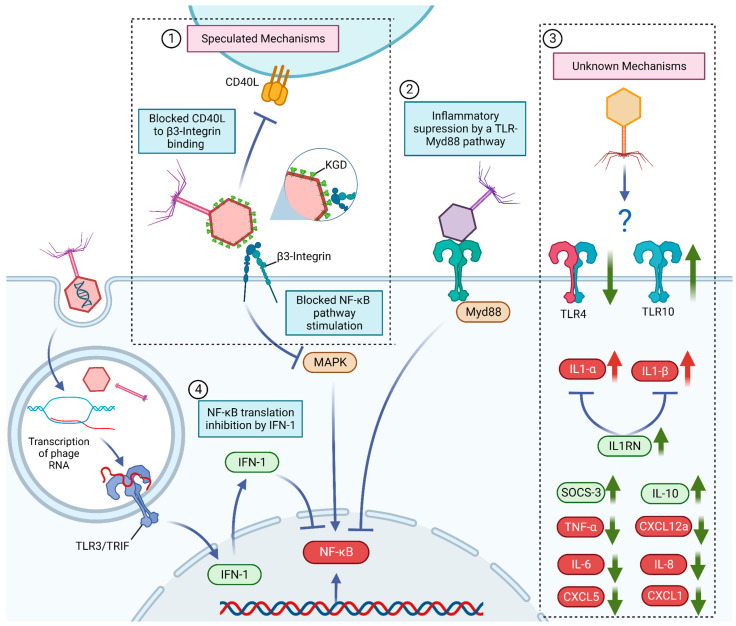
Known, speculated and undiscovered mechanisms behind bacteriophage anti-inflammatory activity. ① Blocking of β3-integrin-NF-κB and CD40L-β3-integrin pathways by KGD-presenting bacteriophages. ② NF-κB inhibition by a TLR-Myd88 pathway. ③ Unknown mechanisms behind a bacteriophage’s anti-inflammatory effects, namely TLR4 decrease, TLR10 increase, decrease in pro-inflammatory cytokines and chemokines TNF-α, CXCL12a, IL-6, IL-8, CXCL5 and CXCL1; increase in the anti-inflammatory cytokines and chemokines IL-10 and SOCS-3; increase in the IL-1 downregulator IL1RN. ④ Inhibition of NF-κB translation by phage-mediated IFN-1 production though the stimulation of TLR3/TRIF by the virus RNA, transcribed inside the cell.

**Figure 2 microorganisms-11-02222-f002:**
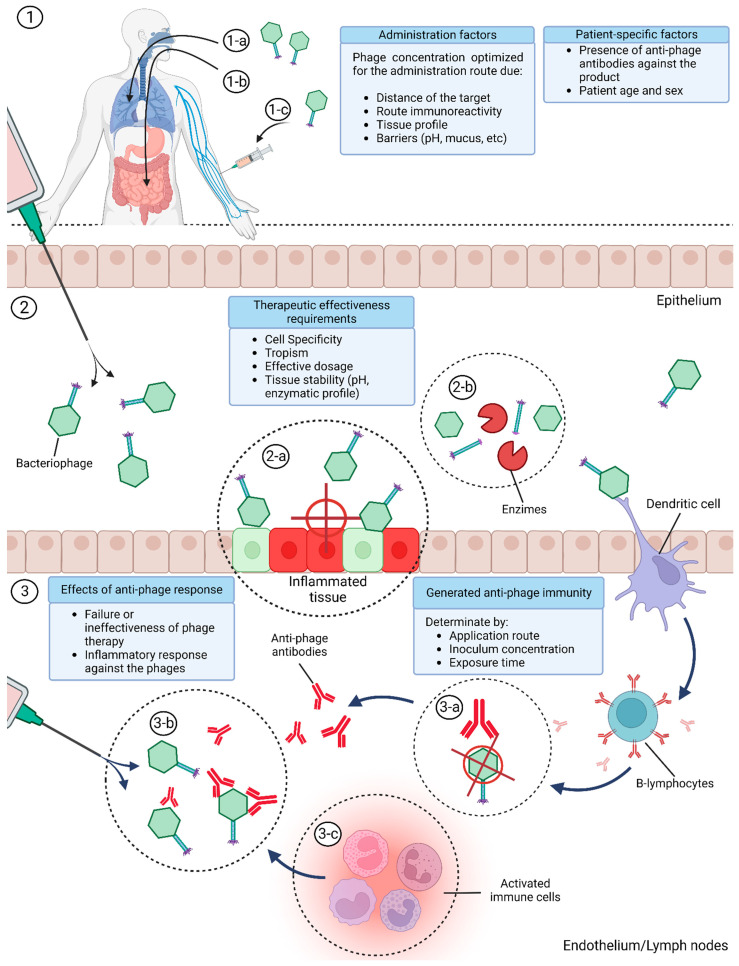
Necessary considerations for the use of bacteriophages as clinical immune modulators for a more effective therapy. ① Conditions between the inoculation by nasal (1-a), oral (1-b) or intravenous route (1-c) and the specific profile of the patient. ② Requirements between the phage-product and the targeted inflamed tissue. (2-a) Specific tropism and effective dosage to the targeted tissue. (2-b) Stability against enzymatic/environmental hazards that may compromise the phage integrity. ③ Immunological consequences of an improper product’s pharmacokinetics. (3-a) Generated adaptive immunity against the phage product. (3-b) Hindrance of the therapy and prevention of future therapies with the same phage product due to the presence of anti-phage antibodies. (3-c) Pro-inflammatory signal due to the recognition of the phage product as an invading body resulting from anti-phage antibody activation.

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
