# Peer review of "Bacteriophages as Potential Clinical Immune Modulators"

_microorganisms, 2023, doi:10.3390/microorganisms11092222_

Round 1
Reviewer 1 Report
The manuscript offers a comprehensive analysis of the nuanced relationship between bacteriophages and inflammation. It elaborates on both their anti-inflammatory and pro-inflammatory effects. Relying on a vast array of sources, the manuscript presents a well-balanced perspective on the role of bacteriophages in modulating immune responses. It elucidates the various mechanisms guiding these interactions, accentuates the different outcomes triggered by distinct phage strains, and ponders the potential variables that might influence these effects. Generally speaking, the work provides a deep dive into a specialized domain of bacteriophages, evident of rigorous research and expertise.
However, its technical language, although indicative of its academic rigor, might pose comprehension challenges for a wider audience. To make it more universally appealing, there's a need to simplify and clarify the content, ensure uniformity in terms, and better the narrative's continuity.
Please check the comments and suggestions below:
Page 1
- One the -> one of the
Page 2
- Due their capacity -> due to their capacity
- Phagetherapy -> phage therapy (throughout the manuscript)
- Antiinflammatory -> anti-inflammatory (throughout the manuscript)
- Proinflammatory cytokines, chemokines -> pro-inflammatory cytokines and chemokines
- inflammation associated genes -> inflammation-associated genes
- T cell activation in in vitro and in vivo models" -> T cell activation in both in vitro and in vivo models
- properties of phages were strong -> anti-inflammatory properties of phages were significant
Page 3
- LPS -> lipopolysaccharide (LPS)
- IL-1a -> IL-1α: It seems like a typographical error for the Greek letter alpha.
- GROa/CXCL1 -> GROα/CXCL1: Again, the correction for the Greek letter alpha.
- Phage Anti-inflammatory Activity -> Phage Anti-Inflammatory Activity
- phage inoculums -> phage inocula: The plural form of "inoculum" is "inocula"
- auto-immune -> autoimmune
- Besides LPS Entrapment -> Beyond LPS Entrapment
- such indirectly modulation -> such indirect modulation
Page 4
- IL-1alpha -> IL-1α
- phagecell -> phage-cell
- inflammations -> inflammation
Page 5
- with animal cells metabolism -> with animal cell metabolism
- phage-eukaryotic -> phage-eukaryote
- a seemingly bacteria exclusive virus -> a virus seemingly exclusive to bacteria
- interference of the phage proteins -> interference by phage proteins
- KGD (Lys-Gly-Asp) compound -> KGD (Lys-Gly-Asp) motif
- RNA by TLR3 -> RNA through TLR3
- antibacterial-direct inflammation -> direct antibacterial inflammation
- interleukin-12 (Il-12) -> interleukin-12 (IL-12)
- were found after phage application, -> were observed post-phage application
Page 7
- IL1A, IL1B -> IL-1A, IL-1B
- IL-1a, IL-1b -> IL-1A, IL-1B
- interferon-g (IFN-c) -> interferon-gamma (IFN-γ)
- TNF-a -> TNF-α
- IL1RN -> IL-1RN
- viral tilter -> viral titer
- in in vivo phage-therapy -> in in-vivo phage therapy
Page 8
- effect, once -> effect because
- phage tilters -> phage titers
- were found to dictated phage -> were found to dictate phage
- in Sun et al report -> in the report by Sun et al.
- Zhang et al. study -> study by Zhang et al.
- phage-product as an invading body -> phage product as an invading body
- the lack of consent in administration methods -> the lack of consensus on administration methods
- phage pharmacology remains an undiscovered field -> phage pharmacology is still a largely unexplored field. Or… Please re-write the sentence.
- phage immunomodulatory properties as a new therapeutic line -> the immunomodulatory properties of phages as a new therapeutic avenue
The technical language, although indicative of its academic rigor, might pose comprehension challenges for a wider audience. To make it more universally appealing, there's a need to simplify and clarify the content, ensure uniformity in terms, and better the narrative's continuity.
Author Response
Dear revisor,
We appreciate your time and review of the "Bacteriophages as Potential Clinical Immune Modulators".
Below are the requested changes to the document and the response to your comments.
#Reviwer 1
Comments and Suggestions for Authors
Page 1
- One the -> one of the
Response: Thank you for the observation. The suggestion was accepted and the text was modified on the abstract.
Page 2
- Due their capacity -> due to their capacity
Response: Thank you for the observation, the suggestion was accepted and the text was modified on the first paragraph of section 1, introduction, page 1.
- Phagetherapy -> phage therapy (throughout the manuscript)
Response: Thank you for the observation, the suggestion was accepted and the text was modified on the abstract, page 1 and 2 of the first and second paragraph of section 1, page 7 on the first paragraph of section 7, page 8 and 9 on the first and second paragraph of section 8, page 9 on the first paragraph of section 9.
- Antiinflammatory -> anti-inflammatory (throughout the manuscript)
Response: Thank you for the observation, the suggestion was accepted and the text was modified on page 5, on the title of section 6.
- Proinflammatory cytokines, chemokines -> pro-inflammatory cytokines and chemokines
Response: Thank you for the observation, the suggestions were accepted and the text was modified on the second paragraph of section 1, page 2.
- inflammation associated genes -> inflammation-associated genes
Response: Thank you for the observation, the suggestion was accepted and the text was modified on the second paragraph of section 1, page 2.
- T cell activation in in vitro and in vivo models" -> T cell activation in both in vitro and in vivo models
Response: Thank you for the observation, the suggestion was accepted and the text was modified on the first paragraph of section 2, page 2.
- properties of phages were strong -> anti-inflammatory properties of phages were significant
Response: Thank you for the suggestion, the suggestion was accepted and the text was modified on the first paragraph of section 2, page 2.
Page 3
- LPS -> lipopolysaccharide (LPS)
Response: Thank you for the observation, the suggestion was accepted and the text was modified on the second paragraph of section 2, page 3.
- IL-1a -> IL-1α: It seems like a typographical error for the Greek letter alpha.
Response: Thank you for the observation, the suggestion was accepted and the text was modified on the second paragraph of section 2, page 3. Figure 1, on page page 6, also had IL-1A modified to IL-1α.
- GROa/CXCL1 -> GROα/CXCL1: Again, the correction for the Greek letter alpha.
Response: Thank you for the observation, the suggestion was accepted and the text was modified on the second paragraph of section 2, page 3
- Phage Anti-inflammatory Activity -> Phage Anti-Inflammatory Activity
Response: Thank you for the observation, the suggestion was accepted and the text was modified on the title of section 3, page 3.
- phage inoculums -> phage inocula: The plural form of "inoculum" is "inocula"
Response: Thank you for the observation, the suggestion was accepted and the text was modified on section 3 on page 3 and section 4 on page 4.
- auto-immune -> autoimmune
Response: Thank you for the observation, the suggestion was accepted and the text was modified on section 3 on page 3 and on the second paragraph of section 9, page 11.
- Besides LPS Entrapment -> Beyond LPS Entrapment
Response: Thank you for the suggestion, the suggestion was accepted and the text was modified on the title of section 4, page 3.
- such indirectly modulation -> such indirect modulation
Response: Thank you for the observation, the suggestion was accepted and the text was modified on section 5, page 4.
Page 4
- IL-1alpha -> IL-1α
Response: Thank you for the observation, the suggestion was accepted and the text was modified on the first paragraph section 4 on page 4.
- phagecell -> phage-cell
Response: Thank you for the observation, the suggestion was accepted and the text was modified on section 4, page 4.
- inflammations -> inflammation
Response: Thank you for the observation, the suggestion was accepted and the text was modified on section 4, page 4.
Page 5
- with animal cells metabolism -> with animal cell metabolism
Response: Thank you for the observation, the suggestion was accepted and the text was modified on the first paragraph of section 6, page 5.
- phage-eukaryotic -> phage-eukaryote
Response: Thank you for the observation, the suggestion was accepted and the text was modified on the first paragraph of section 6, page 5.
- a seemingly bacteria exclusive virus -> a virus seemingly exclusive to bacteria
Response: Thank you for the suggestion, the suggestion was accepted and the text was modified on the first paragraph of section 6, page 5.
- interference of the phage proteins -> interference by phage proteins
Response: Thank you for the suggestion, the suggestion was accepted and the text was modified on the first paragraph of section 6, page 5.
- KGD (Lys-Gly-Asp) compound -> KGD (Lys-Gly-Asp) motif
Response: Thank you for the suggestion, the suggestion was accepted and the text was modified on the first paragraph of section 6, page 5.
- RNA by TLR3 -> RNA through TLR3
Response: Thank you for the suggestion, the suggestion was accepted and the text was modified on section 4, page 4.
- antibacterial-direct inflammation -> direct antibacterial inflammation
Response: Thank you for the suggestion, however we found that “antibacterial directed inflammation” is more appropriate.
- interleukin-12 (Il-12) -> interleukin-12 (IL-12)
Response: Thank you for the observation, the suggestion was accepted and the text was modified on the second paragraph of section 6, page 5.
- were found after phage application, -> were observed post-phage application
Response: Thank you for the suggestion, the suggestion was accepted and the text was modified on the third paragraph of section 6, page 6.
Page 7
- IL1A, IL1B -> IL-1A, IL-1B
Response: Thank you for the suggestion, however IL-1A was changed to IL-1α and IL-1B was changed to IL-1β, for the same format is used along the text. Figure 1, on page page 6, also had IL-1A modified to IL-1α and IL-1B changed to IL-1β.
- IL-1a, IL-1b -> IL-1A, IL-1B
Response: Thank you for the suggestion, the same modifications were applied followed by the item above.
- interferon-g (IFN-c) -> interferon-gamma (IFN-γ)
Response: Thank you for the observation, the suggestion was accepted and the text was modified on section 7, page 7. However only the abbreviation was employed, since interferons were already indicated before as “IL” .
- TNF-a -> TNF-α
Response: Thank you for the observation, the suggestion was accepted and the text was modified on section 7 on page 7 and on section 2 on page 2.
- IL1RN -> IL-1RN
Response: Thank you for the observation, the suggestion was accepted and the text was modified on section 7 on page 7.
- viral tilter -> viral titer
Response: Thank you for the observation, the suggestion was accepted and the text was modified on section 8, page 7.
- in in vivo phage-therapy -> in in-vivo phage therapy
Response: Thank you for the observation, the text was simply rewritten as “in vivo phage-therapy” on section 8, page 8.
Page 8
- effect, once -> effect because
Response: Thank you for the suggestion, the text was instead rewritten as “effect since”.
- phage tilters -> phage titers
Response: Thank you for the observation, the suggestion was accepted and the text was modified on section 8, page 8.
- were found to dictated phage -> were found to dictate phage
Response: Thank you for the observation, the suggestion was accepted and the text was modified on section 8, page 8.
- in Sun et al report -> in the report by Sun et al.
Response: Thank you for the observation, the suggestion was accepted and the text was modified on section 8, page 8.
- Zhang et al. study -> study by Zhang et al.
Response: Thank you for the suggestion, the text was instead rewritten as “in the report by Zhang et al.” on section 8, page 8..
- phage-product as an invading body -> phage product as an invading body
Response: Thank you for the observation, the suggestion was accepted and the text was modified on the legend of Figure 2, page 11.
- the lack of consent in administration methods -> the lack of consensus on administration methods
Response: Thank you for the observation, the suggestion was accepted and the text was modified on section 9, page 11.
- phage pharmacology remains an undiscovered field -> phage pharmacology is still a largely unexplored field. Or… Please re-write the sentence.
Response: Thank you for the suggestion, the text was instead rewritten as “phage pharmacology is still a largely unexplored field” on the second paragraph of section 9, page 11.
- phage immunomodulatory properties as a new therapeutic line -> the immunomodulatory properties of phages as a new therapeutic avenue
Response: Thank you for the suggestion, the suggestion was accepted and the text was modified on the second paragraph of section 9, page 11.
.
- However, its technical language, although indicative of its academic rigor, might pose comprehension challenges for a wider audience. To make it more universally appealing, there's a need to simplify and clarify the content, ensure uniformity in terms, and better the narrative's continuity.
Response: Thank you for your suggestion. The language was revised in search of a comprehensive text without losing scientific and technical rigor for the readership.
Reviewer 2 Report
The review "Bacteriophages as Potential Clinical Immune Modulators" is very interesting. It focuses on phage therapy and its relationship to the modulation of the immune response.
I have some comments that I share with the authors:
Minor fixes:
Throughout the text and in Figure 1, it is crucial to detail the meaning of the acronyms. Although these molecules are widely recognized in the field, it is essential to consider those readers who show interest in their research without being experts in the area.
For example, IL…, CXCL.., LPS, TNF, NF-KB, HSV-1, etc.
Optional
Perhaps they would like to add a few other sections to their review. For example:
-Interactions between bacteriophages, bacteria, and the immune system in the context of the microbiome.
-Therapeutic potential of microbiome modulation with bacteriophages.
Author Response
Dear revisor,
We appreciate your time and review.
Below are the requested changes to the document and the response to your comments.
#Reviwer 2
Comments and Suggestions for Authors
I have some comments that I share with the authors:
Minor fixes:
Throughout the text and in Figure 1, it is crucial to detail the meaning of the acronyms. Although these molecules are widely recognized in the field, it is essential to consider those readers who show interest in their research without being experts in the area.
For example, IL…, CXCL.., LPS, TNF, NF-KB, HSV-1, etc.
Response: Thank you for the suggestion, the text was modified to contain the meaning of each acronym in order of its first mention.
The full form of the acronyms was added in the first and second paragraph on section 2 on page 3, in 3 section on page 3 and in section 7 on page 7.
Optional
Perhaps they would like to add a few other sections to their review. For example:
-Interactions between bacteriophages, bacteria, and the immune system in the context of the microbiome.
-Therapeutic potential of microbiome modulation with bacteriophages.
Response: Thank you for your suggestions.
Topics involving interactions between microbiome phages and human cells are little explored and an important area of knowledge to be elucidated. However, in the present review, our main objective was to deepen and explore the direct interactions between phages and eukaryotic cells, addressing "mechanisms for which bacteriophages may interact with immune cells, analyzing potential clinical applications and obstacles presented in the use of bacteriophages as anti-inflammatory tools".
Round 2
Reviewer 1 Report
The manuscript was revised well.
I have one minor suggestion regarding authorship. While you've noted that first authorship is shared among all authors by Line 365, I believe it would be beneficial to include this information on the first page alongside the corresponding author details for the sake of clarity and visibility. This will help ensure that the equal contributions of all authors are recognized from the outset, rather than requiring the reader to reach Line 365 for this information.
Thank you.